# Development of a new method to promote tooth extraction socket healing through immediate transplantation of extracted dental pulp tissue - An in vivo study

Kohei Ijichi, Reiko Tokuyama-Toda[ID]*, Mai Takeshita-Umehara, Toshio Yudo, Yusuke Takebe, Kazuhito Satomura

Department of Oral Medicine and Stomatology, Tsurumi University School of Dental Medicine, Yokohama, Japan

* tokuyama-r@tsurumi-u.ac.jp

## Abstract

In tooth extraction sockets, natural healing is usually expected; however, early healing is advantageous for subsequent treatment and leads to improvement in the patient's quality of life. In recent years, it has been found that dental pulp tissue contains several dental pulp stem cells. The main objective of this study was to verify whether immediate transplantation of dental pulp tissue without any culture process into the extraction socket would promote healing of the socket. To verify this, the rat's bilateral maxillary second molars were extracted simultaneously, and dental pulp tissue was collected from the extracted teeth. The right extraction socket was immediately transplanted with dental pulp tissue, and the left extraction socket was used as a control. Both sockets were then covered with immediate curing resin. Micro-CT was performed over time, up to 28 days after tooth extraction, bone mineral content was measured, and histological examination was performed. As a result, inflammatory bone resorption had a normal course in both groups after tooth extraction, but bone resorption tended to be suppressed on the 3 days after tooth extraction in the transplant group. In the transplant group, compared to the control group, the radiopaque area increased from the 5 days after tooth extraction onward, and a significant difference was observed on the 7 days after tooth extraction. Furthermore, bone mineral content measurements suggested that bone maturation progressed from earlier in the transplant group, and histological examination confirmed that bone healing was promoted in the transplant group. These findings confirmed that immediate transplantation of dental pulp tissue can promote bone healing in the extraction socket and suggested that this method can be useful as a new method to promote healing in the extraction socket.

**Data availability statement:** All relevant data are within the manuscript.

**Funding:** This research was funded by JSPS KAKENHI [grant numbers 23K0938 and 21K19617].

**Competing interests:** The authors declare no conflicts of interest.

## Introduction

Tooth extraction is one of the most prevalent surgical procedures in clinical oral surgery [1–3]. The healing process of the tissue defect created by tooth extraction, known as the extraction socket, can be divided into four main phases [4–6]: (1) the clotting phase, in which the extraction socket is filled with coagulum; (2) the granulation tissue phase, in which the coagulum is replaced by granulation tissue; (3) the callus formation phase, in which osteogenesis is observed around the extraction socket; and (4) the healing phase, in which the wound margin is closed and the alveolar crest is flattened. The healing phase is followed by closure of the wound margin and alveolar apex flattening [4–6]. Immediately after tooth extraction and until a certain degree of healing has occurred, patients experience discomfort due to the inflammatory response. In particular, when a wisdom tooth is extracted, quality of life (QOL) is significantly reduced due to the location and size of the defective tooth [1–3,7]. In addition, in cases of tooth extraction for orthodontic treatment, early treatment is desirable, but orthodontic treatment is usually postponed until the extraction socket has reached a certain stage of healing [8]. Nowadays, when a tooth is extracted in clinical practice, it is common for the extraction socket to undergo natural healing, but if this healing period can be shortened, this will go a long way toward improving the patient's QOL and the early start of subsequent orthodontic treatment. Meanwhile, mesenchymal stem cells (MSCs) derived from bone marrow and adipose tissue have been applied clinically as cell sources for regenerative medicine [9–14], and it has been reported that stem cells also exist in human dental pulp tissue [15–18]. As dental pulp stem cells (DPSCs) have a cell proliferation capacity and multipotency similar to or greater than stem cells derived from other tissues [19–25], various possibilities for clinical application are being investigated [26–28]. Among the tooth extractions carried out in the clinical oral surgery mentioned above, in cases such as wisdom tooth extractions or tooth extractions for orthodontic treatment, the teeth themselves are usually healthy, which means that the dental pulp tissue is also healthy, but they are usually discarded as hospital waste. The purpose of this study was to investigate the healing-promoting effect of transplanting healthy dental pulp tissue from an extracted tooth directly into the extraction socket without a culture process.

## Materials and methods

Our animal care procedures and experimental protocols were approved by the Animal Experiments Ethics Committee of Tsurumi University (authorization number: 23P035). All the methods were carried out in accordance with the relevant guidelines and regulations. The surgical procedures were performed to minimize the suffering of the laboratory animals. The animals were fed a normal diet and were kept on a 12-h light–dark cycle at 22°C. The sample number was elected by power test analysis. Level of significance of 5% and power test of 95% were adopted, and it was suggested four animals per group. Thus, with a possible animal loss, it was used five per period of analysis (S1 Table).

## Collection and immediate transplantation of dental pulp tissue into the tooth extraction socket

The study was conducted on 30 six-week-old male Sprague–Dawley (SD) rats (5 for micro-CT analysis and 25 for histological analysis). The rats were obtained from the Nippon Clare Corporation (Tokyo, Japan). The maxillary bilateral second molars (M2) of the SD rats were extracted simultaneously under a combination anesthetic (0.375 mg/kg of medetomidine, 2.0 mg/kg of midazolam, and 2.5 mg/kg of butorphanol). The two extracted teeth were cut at the cement-enamel junction, and dental pulp tissue was collected from the pulp cavity. The dental pulp tissue collected was immediately transplanted into the extraction socket on the right side, which was designated the dental pulp tissue transplant group (n = 5 at each point). The left side was designated the control group, in which no transplantation was performed and the natural healing process was monitored (n = 5 at each point). Extraction socket surfaces were covered with self-curing adhesive resin cement (Superbond®, Sun Medical Co., Moriyama, Japan) on both sides to protect the extraction sockets from the external environment. These procedures were performed by a single experienced operator (K.I.) under the guidance of an oral surgeon certified by the Japanese Society of Oral and Maxillofacial Surgeons (R.T-T.). The image analysis was conducted by three authors (R.T-T. (the corresponding author), K.I. (the first author), and K.S.).

## Observation of tooth extraction sockets using microcomputed tomography (micro-CT)

The extraction sockets in the rat jaws were imaged in vivo for the longitudinal experiments. The extraction sockets were observed using micro-CT on days 1, 3, 5, 7, 10, 14, 21, and 28 after tooth extraction, and the healing process of the extraction sockets was compared between the two groups based on variations in the radiopaque area ($mm^3$). A high-resolution X-ray CT (inspXio SMX-225CT, Shimadzu, Kyoto, Japan) was used to non-destructively observe the internal structure of maxilla of five Sprague Dawley (SD) rat at each scheduled day. The imaging conditions were as follows: tube voltage, 110 kV; tube current, 70 µA; resolution, 1024 × 1024 pixels; number of scanning views, 1200, and total scanning time, 900 s. Image processing software (TRI/3D-BON-FCS64, RATOC, Tokyo, Japan) was used for 3D image analysis. A previously reported method [29–31] was modified to determine radiolucent and radiopaque areas in tooth extraction sockets. That is, due to the irregular alveolar socket morphology, the region of interest (ROI) was standardized as a cube enclosed by the determined plane, which based on a straight line passing through the center of the distal buccal roots of the bilateral maxillary first molars and the tooth axis of the distal buccal roots of the bilateral maxillary first molars. The above plane was determined on the 5th day when the radiolucent area expanded due to bone resorption caused by inflammation after tooth extraction, and the range of the extraction socket to be measured was determined to be the range from 15–20 sections to 90–95 sections from the set plane (Fig 1). In this manner, based on the images obtained under the same conditions at all time points, the radiopaque area and the bone mineral content was measured and compared between the two groups (n = 5 at each scheduled day).

The micro-CT images were used to analyze changes the volume of the radiopaque area in the extraction socket over time. In addition, the volume change was calculated by comparing it with the previous measurement, and the rate of change was calculated to compare the changes in the radiopaque area between the two groups. In addition, the bone mineral content of the increased radiopaque areas, which is believed to reflect new bone formation, was measured. Then, at each time point, the volume of the area observed at 1,275 mg/cm³ or more (value between yellow and green in the color bar), which is considered to have progressed to a certain degree of bone maturation, was compared between the two groups.

## Histological study

On days 7, 10, 14, 21, and 28 after tooth extraction, 25 animals (n = 5 per group) were euthanized by anesthesia overdose (pentobarbital sodium, 100 mg/kg). The maxillary bone, including the bilateral extraction socket, was

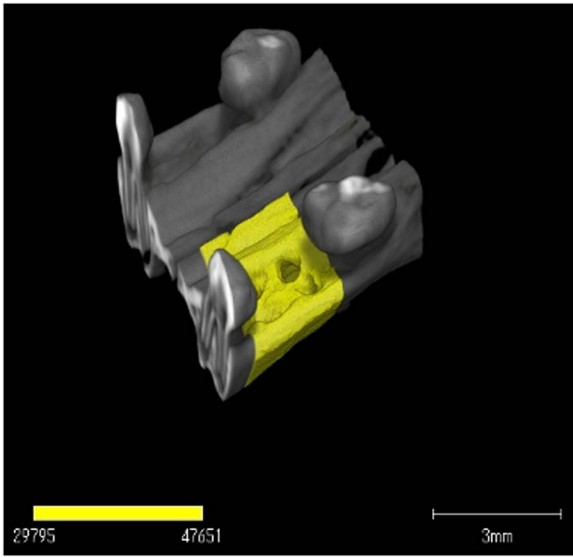

**Fig 1. Micro-CT evaluation of extraction socket.** Micro-CT analysis using CT Analyzer software. The area outlined in yellow indicates the region of interest (ROI) placed to encompass the extraction socket during bone healing.

removed (n = 5 at each point); fixed in 4% paraformaldehyde phosphate buffer (4% PFA; Fujifilm Wako Pure Chemicals Corporation, Osaka, Japan), and then decalcified with 17% ethylene diamine tetraacetic acid (Osteosoft˚; Sigma Aldrich Japan, Tokyo, Japan) for 4 weeks. The fragments were embedded in paraffin and cut to a thickness of 6 µm. The paraffin sections were deparaffinized, hydrated, and stained with hematoxylin and eosin before histological observations.

## Statistical analysis

The results are presented as means ± standard errors. All data indicated at least three individual experiments and were analyzed using the Mann–Whitney U-test. Differences were considered significant at $p < 0.05$.

## Results

### Changes in the extraction socket over time, as seen in micro-CT images

Fig 2A shows sequential micro-CT images of the same area in the frontal region. The right side shows the dental pulp tissue transplant group, and the left side shows the control group. In both groups, an increase in the radiopaque area in the extraction socket was observed over time from the fifth day after tooth extraction. In particular, on days 10 and 14, alveolar crest flattening was observed earlier in the dental pulp tissue transplant group than in the control group. However, in the micro-CT images taken on days 21 and 28, almost no visual difference in the radiopaque area was observed between the transplant and control groups. Therefore, the changes in the radiopaque area were compared between the two groups are presented in Fig 2B, the amount of volume change are presented in Fig 2C, and the rate of volume change are presented in Fig 2D. In addition, these numerical results are shown in Table 1. As a result, a slight decrease in the volume of the radiopaque area was observed 3 days after tooth extraction in both groups (control group −6.428%; dental pulp tissue transplant group −7.503%), indicating temporary bone resorption associated with a normal inflammatory response after tooth extraction. Five days after tooth extraction, bone resorption was still progressing in

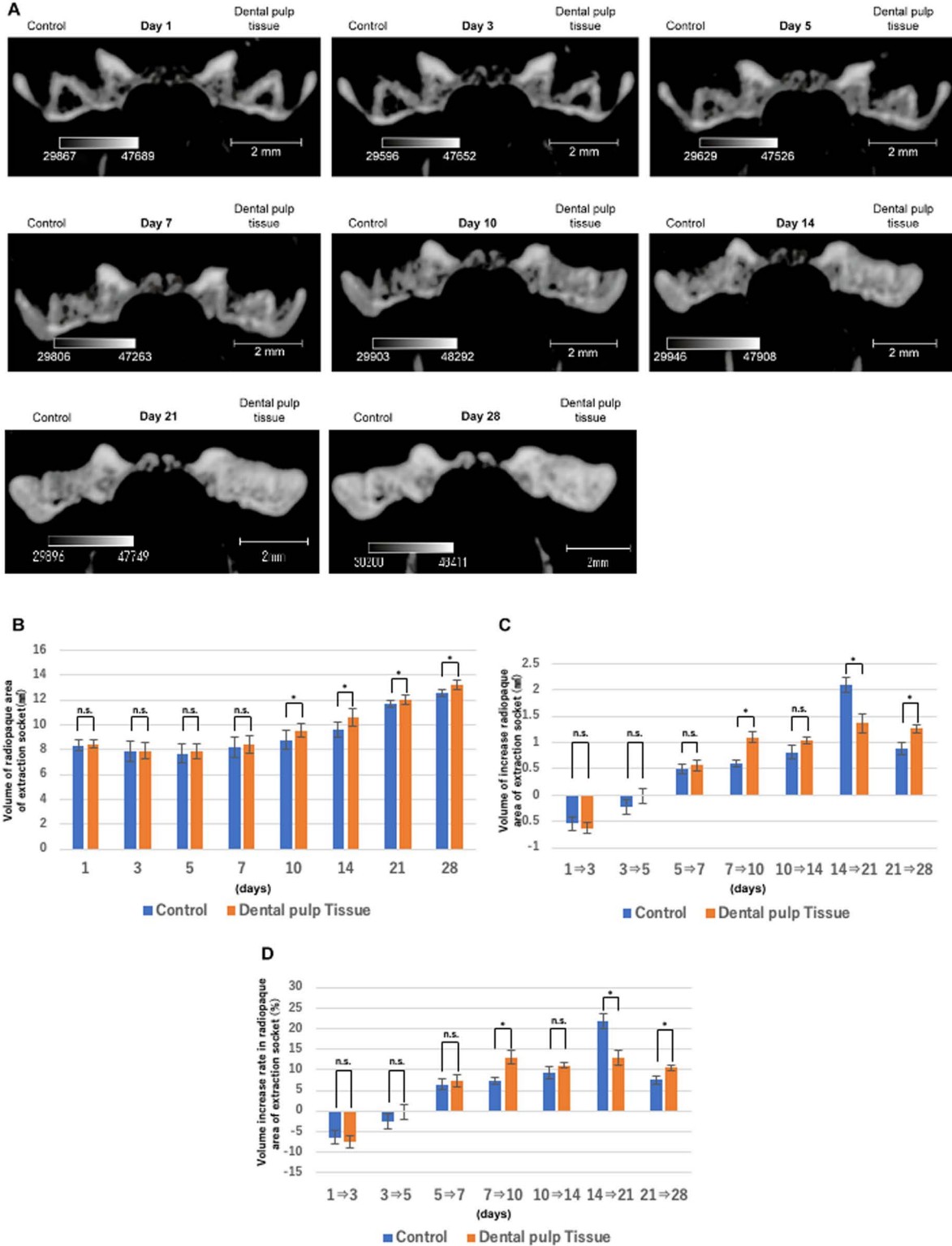

**Fig 2. Temporal changes in the extraction socket. (A)** Frontal section micro-CT images including bilateral tooth extraction sockets. **(B)** Volume of the radiopaque area. When the volume of the radiopaque area was compared between the two groups, a greater radiopaque area was observed in the transplanted group on days 10 and 14 (day 10: control group 8.7888 mm³; dental pulp tissue transplant group 9.5508 mm³, day 14: control group

9.6052 mm³; dental pulp tissue transplant group 10.5988 mm³). **(C)** Amount of increase in the volume of the radiopaque area over time and (D) its rate of change. In the control group, the radiopaque area decreased until day 5; in the transplant group, this period was reduced to day 3. The peak increase in the radiopaque area was observed between days 14 and 21 in the control group (21.809%) and between days 7 and 10 in the transplant group (13.062%) (n = 5).

**Table 1. Numerical results of amount of increase in the volume of the radiopaque area over time and its rate of change.**

|  | Volume of increase radiopaque area of extraction socket (mm³) | | | Volume increase rate in radiopaque area of extraction socket (%) | | |
|---|---|---|---|---|---|---|
|  | Control | Dental pulp Tissue | *P*- value | Control | Dental pulp Tissue | *P*-value |
| 1⇒3 | −0.5422 ± 0.137 | −0.626 ± 0.150 | 0.1263 | −6.4284 ± 1.976 | −7.50347 ± 1.820 | 0.1256 |
| 3⇒5 | −0.2156 ± 0.182 | −0.0136 ± 0.184 | 0.0791 | −2.4788 ± 2.081 | −0.17232 ± 2.187 | 0.0766 |
| 5⇒7 | 0.4906 ± 0.124 | 0.5688 ± 0.145 | 0.0898 | 6.3704 ± 1.558 | 7.21955 ± 1.918 | 0.0927 |
| 7⇒10 | 0.597 ± 0.0791 | 1.1034 ± 0.148 | 0.00637 | 7.2877 ± 0.843 | 13.0620 ± 1.895 | 0.00693 |
| 10⇒14 | 0.8164 ± 0.165 | 1.048 ± 0.086 | 0.0625 | 9.2890 ± 1.595 | 10.9729 ± 1.079 | 0.231 |
| 14⇒21 | 2.0948 ± 0.140 | 1.3692 ± 0.175 | 0.00235 | 21.809 ± 1.787 | 12.9184 ± 1.892 | 0.00106 |
| 21⇒28 | 0.888 ± 0.109 | 1.26 ± 0.0783 | 0.0199 | 7.5897 ± 0.987 | 10.5280 ± 0.702 | 0.0223 |

The increase and percentage change in radiopaque area on each scheduled day will be expressed as numerical results along with P values.

the control group (−2.478%), whereas it had stopped in the dental pulp tissue transplant group. Meanwhile, an increase in the radiopaque area was observed in both groups from day 7 onward. However, the increase was greatest from day 7 to day 10 in the dental pulp tissue transplant group (13.062%) and from day 14 to day 21 in the control group (21.809%), indicating that the radiopaque area increased earlier in the dental pulp tissue transplant group. Fig 3A shows the bone mineral content in the tooth extraction sockets in the two groups. In both groups, an increase in bone mineral content was observed over time. Then, the volume of the area showing a score of 1275 mg/cm³ or more in both groups are presented in Fig 3B. As a result, no differences were observed between the two groups until day 14, but in the dental pulp tissue transplant group, bone mineral content was generally higher on day 21(control group 3.934 mg; dental pulp tissue transplant group 4.475 mg), although this was not significant, and bone mineral content was significantly higher on day 28 (control group 5.695 mg; dental pulp tissue transplant group 6.584 mg). Moreover, many areas with high bone mineral content were observed near the alveolar crest in the dental pulp tissue transplant group from day 14 onward.

## Histological examination of tooth extraction sockets

In the micro-CT images described above, the healing process of the tooth extraction sockets on days 7, 10, 14, 21, and 28, when the radiopaque areas were increasing, was examined histologically (Figs 4–8). On day 7 (Fig 4), the tooth extraction sockets in both groups were filled with blood clots (arrows) and granulation tissue composed of inflammatory cells and fibroblasts (arrowheads). Furthermore, new bone formation (*) was observed around the tooth extraction socket in the dental pulp tissue transplant group. On day 10 (Fig 5), the formation of immature new bone was progressing in the tooth extraction socket in both groups, with more new bone formation observed in the dental pulp tissue transplant group. On day 14 (Fig 6), new bone formation had progressed further in both groups compared to day 10, with more new bone formation in the dental pulp tissue transplant group. On day 21 (Fig 7) and day 28 (Fig 8), healing had progressed in both groups. Compared with the control group, the dental pulp tissue transplant group showed a new flattened bone surface (Fig 8). These findings were consistent with those of micro-CT images.

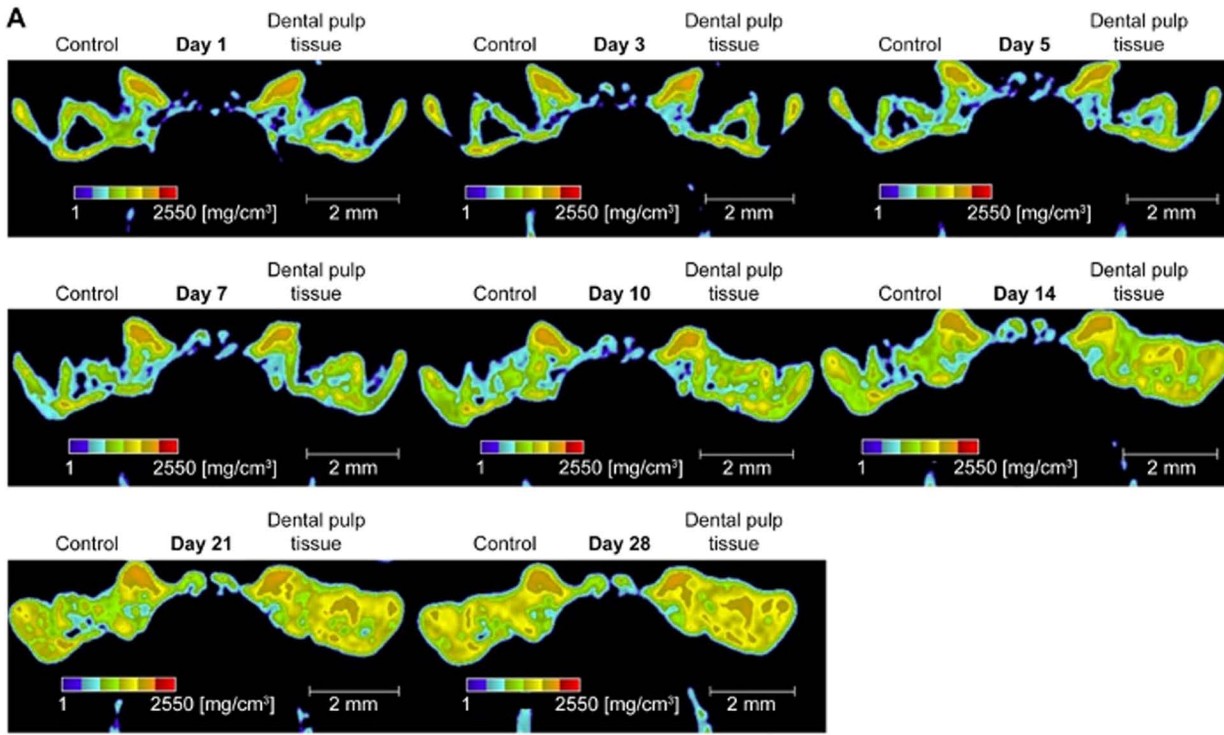

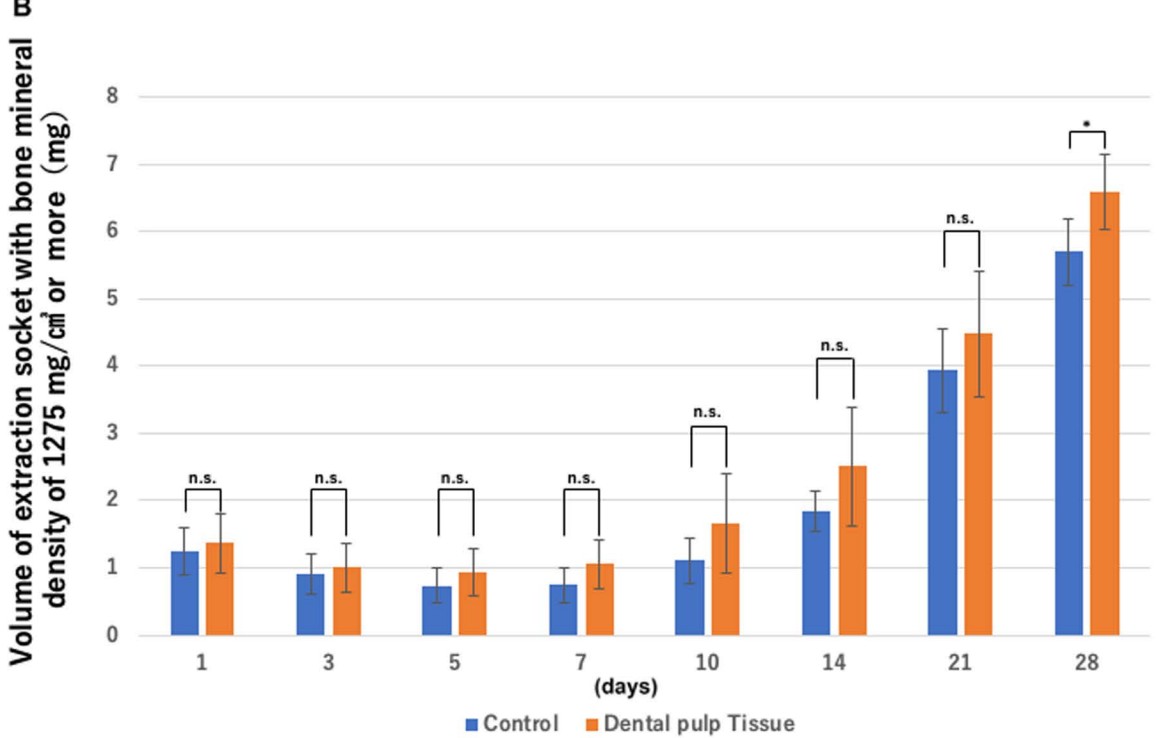

**Fig 3. Changes in bone mineral content over time in extraction sockets. (A)** Measurement of bone mineral content using micro-CT images. **(B)** Volume of areas with a bone mineral content of 1275 mg/cm³ or more. On day 28, there was a significant increase in the amount of mature new bone in the transplant group (control group 5.695 mg; dental pulp tissue transplant group 6.548 mg).

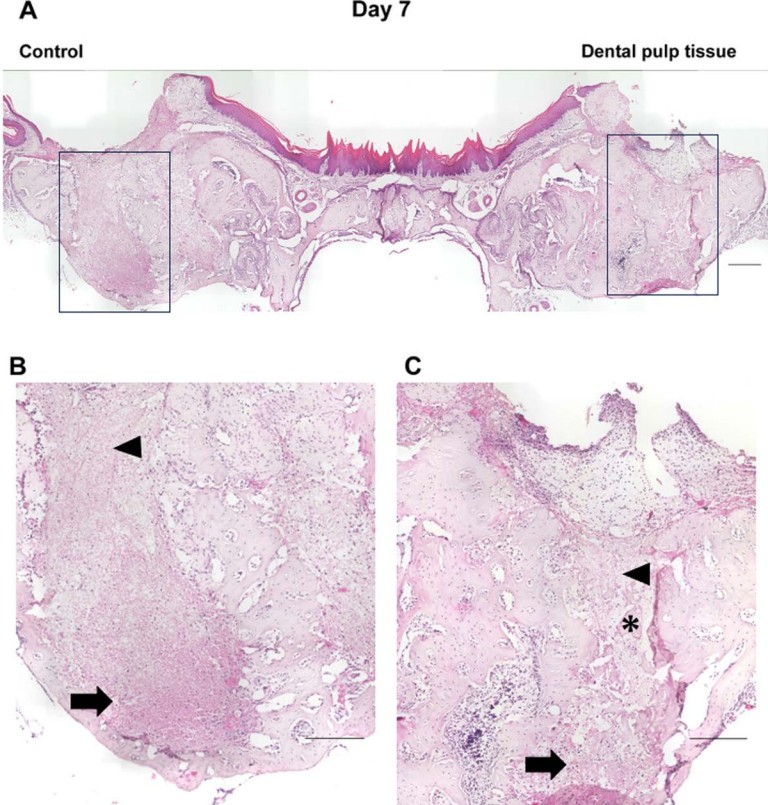

**Fig 4. Histological findings of the extraction socket (day 7). (A)** Overview. **(B)** Control group, **(C)** transplanted group. The extraction sockets in both groups were filled with blood clots (arrows) and granulation tissue composed of inflammatory cells (neutrophils and macrophages) and fibroblasts (arrowheads). Moreover, new bone formation (*) was observed in the transplant group. Scale bars; (A) 300 µm (B, C) 200 µm.

## Discussion

In this study, we focused on healthy dental pulp tissue from extracted teeth that would normally be discarded and investigated the possibility of using dental pulp tissue to simply and effectively promote tooth extraction socket healing. Then, healing promoted in the extraction socket of the dental pulp tissue transplant group was compared to the control group, which followed the natural healing process. As a result, the healing of the tooth extraction socket was promoted earlier in the dental pulp tissue transplant group compared to the control group. From the micro-CT results, bone resorption after tooth extraction, which continues until the 5th day in natural healing, was suppressed by the 3rd day in the transplant group. In addition, the peak of bone formation was observed earlier, from the 7th to 10th day in the transplant group, compared to the 10th to 14th day in the control group. Furthermore, it was observed that the newly formed bone had a higher bone mineral content earlier in the transplant group. Similar to the CT findings, histological findings also showed that bone maturation progressed earlier in the transplant group than in the control group. The alveolar bone healing in the control group supported previous studies [29,32–35], and there were no problems with the healing process in this study. Based on these findings, direct transplantation of extracted dental pulp tissue may promote healing of the extraction socket.

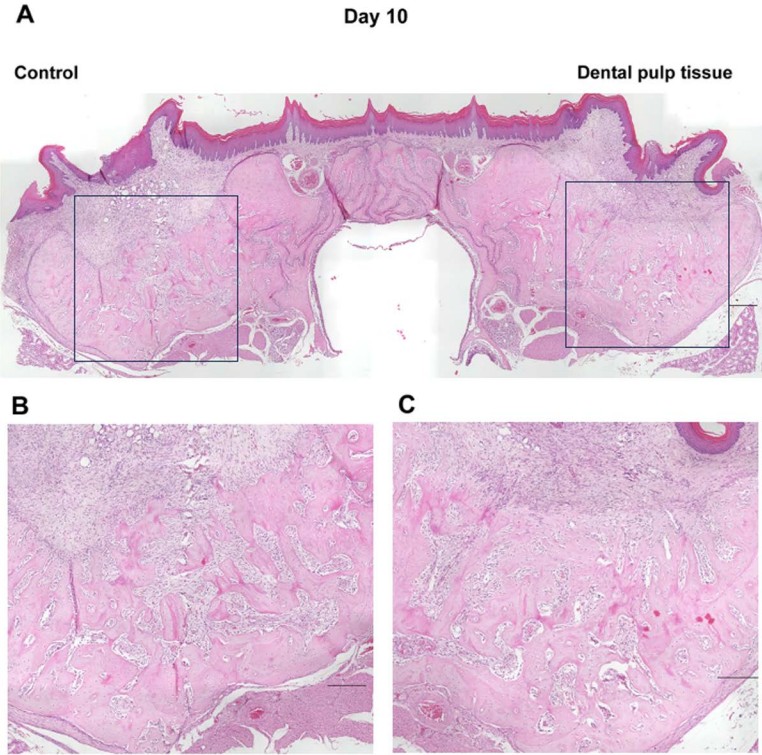

**Fig 5. Histological findings of the extraction socket (day 10). (A)** Overview. **(B)** Control group. **(C)** transplanted group. The formation of immature new bone was progressing in both groups. More new bone formation was observed in the transplant group. Scale bars; (A) 300 μm (B, C) 200 μm.

As for the mechanism, considering the findings observed on the micro-CT images shortly after tooth extraction, it is believed that the transplanted dental pulp tissue not only promoted the formation of new bone in the alveolar bone of the host extraction socket but also exerted an anti-inflammatory effect, which led to an early interruption of the resorption of the alveolar bone caused by the inflammatory response, resulting in the early formation of new bone. Previous studies have shown that prolonged inflammation after tooth extraction delays the healing of the alveolar bone [36–39], and these findings suggest that earlier suppression of the inflammatory response will lead to promoted healing of the tooth extraction socket. Furthermore, considering the early flattening of the alveolar crest in the transplant group observed in the micro-CT images from day 10, it is possible that in the dental pulp tissue transplant group, new bone was not only generated from the existing bone around the extraction socket, but also around the center of the socket due to the influence of the transplanted dental pulp tissue. Previous reports have shown that applying bio-cement to the extraction socket and maintaining the alveolar crest allows for uniform healing of the alveolar bone [40]. In present study, the transplanted dental pulp tissue also promoted new bone formation near the alveolar crest, which may have contributed to the promotion of healing. In addition, bone resorption in the alveolar septum may be suppressed by the transplantation of pulp tissue into the tooth extraction socket, resulting in differences in the morphology of the healed alveolar bone. As described above, although there may be several possible mechanisms, the results of this study suggest that immediate transplantation of dental pulp tissue may be a useful method to promote the healing of tooth extraction sockets.

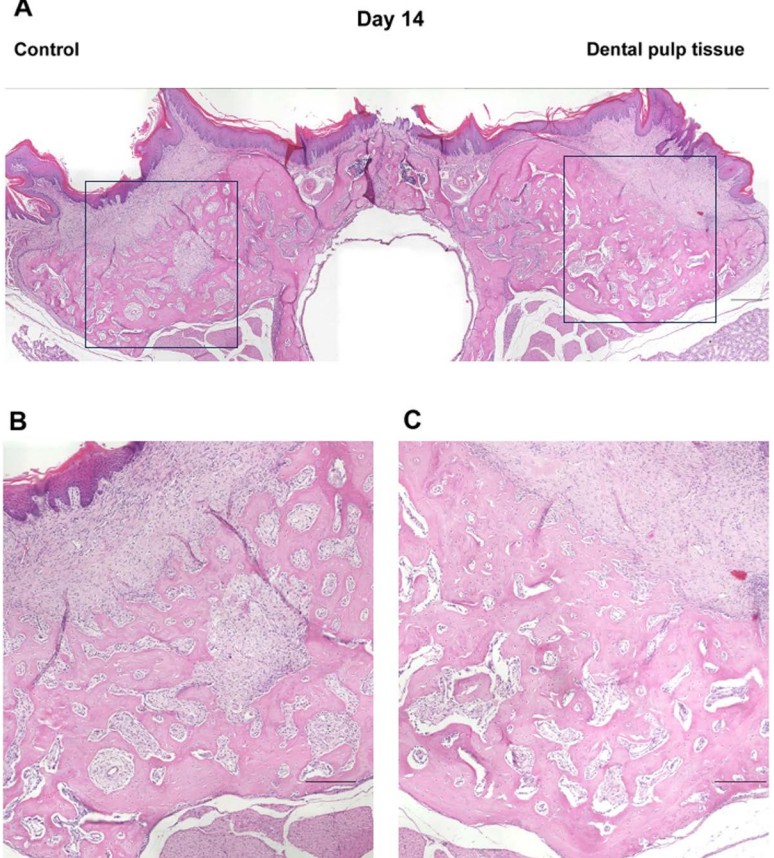

**Fig 6. Histological findings of the extraction socket (day 14). (A)** Overview. **(B)** Control group. **(C)** transplant group. New bone formation progressed more in both groups. More new bone formation was observed in the transplant group. Scale bars; (A) 300 μm (B, C) 200 μm.

Numerous studies have been reported on promoting healing in tooth extraction sockets, whether the subjects were in a normal or delayed healing condition [1–3,30,37,39–47]. In this study, the healing promotion effect was examined in normal tooth extraction sockets. The premise of the results obtained in this study was that the healing process in the control group of tooth extraction sockets followed a normal process in comparison with previous reports [29]. Then, when comparing the results of this study with previous studies that attempted to promote healing of tooth extraction sockets for the same purpose as this study, the results of this study were not significantly superior, but the healing promoting effect was comparable to that of previous studies [30,37,39–47]. It is noteworthy that previous reports used artificial materials [30], required special equipment [43], or transplanted BMSCs that had undergone a culture process [18,45], whereas the method used in this study is safe and simple, using autologous tissue as the transplant material without a culture process. Many studies have been conducted using MSCs, including DPSCs [18,43–47]. Many regenerative therapies using DPSCs as a cell source are being tested, largely due to the high potential of DPSCs [48]. Most of these regenerative therapy investigations used cultured MSCs including DPSCs. However, the current preservation procedure (banking system) requires approximately three weeks from cell isolation to cryopreservation [49,50]. Furthermore, long-term culture increases not only costs and labor but also the risk of contamination [51–54]. From this perspective, this method is extremely advantageous as it

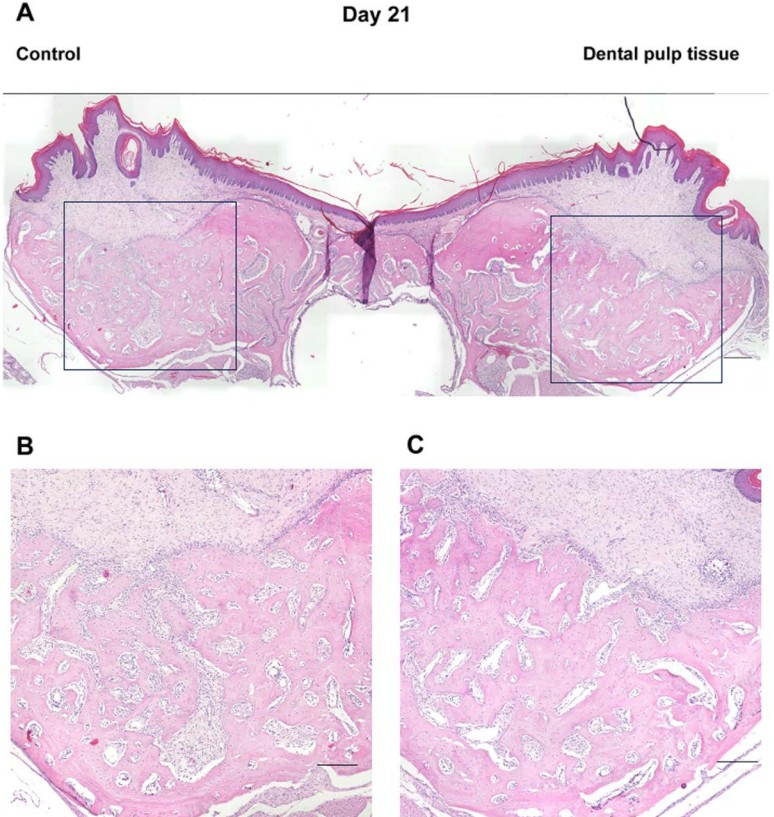

**Fig 7. Histological findings of the extraction socket (day 21). (A)** Overview. **(B)** Control group. **(C)** transplanted group. Healing progressed in both groups. Compared with the control group, the transplant group showed a new flattened bone surface. Scale bars; (A) 300 μm (B, C) 200 μm.

does not require many special procedures or equipment and has a low risk of contamination. In this experiment, we could not clarify the detailed mechanism by which the transplanted dental pulp tissue promotes the healing of the extraction socket. As suggestions for future studies, it will be necessary to elucidate whether the transplanted dental pulp tissue itself differentiates into bone or promotes tissue healing through anti-inflammatory effects or cytokines that induce tissue regeneration in the surrounding tissues. For example, it is planned to perform allogeneic transplants using immunosuppressed animals to clarify the cellular composition of the healing tissue; or to perform transplants in males and females to identify the cellular origin using Y chromosomes; or to identify the cellular origin of the healing tissue by transfecting the green fluorescence protein gene into the transplanted cells to observe how the cells of the transplanted dental pulp tissue interact with the cells of the host bed.

Unfortunately, the results of this study did not show a drastic reduction in the healing period. Therefore, this method needs to be improved before clinical application. For example, studies are planned to use collagen or [55,56] as a scaffolding agent for transplanting dental pulp tissue into the tooth extraction socket or administering melatonin in combination with the transplant to induce anti-inflammatory [57–59] and bone formation effects [56,60,61]. This study confirmed that this method is a safe, simple, and low-cost way of promoting healing and that it does not require any prior preparation, such as a culture process or artificial materials. Therefore, it is considered to apply this method in clinical trials as soon as possible.

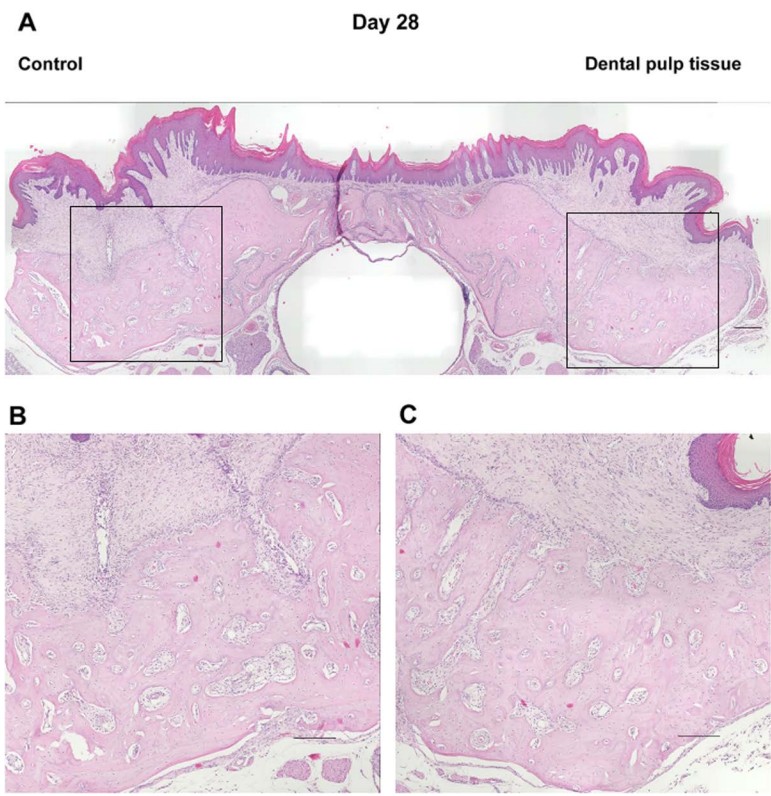

**Fig 8. Histological findings of the extraction socket (day 28). (A)** Overview. **(B)** Control group. **(C)** transplanted group. More new bone was formed in the transplant group than in the control group, and bone flattening in the transplant group. Scale bars; (A) 300 μm (B, C) 200 μm.

## Conclusion

The results suggest that immediate transplantation of dental pulp tissue into tooth extraction sockets promotes alveolar bone healing.

## Supporting information

**S1 Table. Arrive checklist.** This is the arrive checklist about our animal experience.
(DOCX)

## Author contributions

**Conceptualization:** Reiko Tokuyama-Toda, Kazuhito Satomura.

**Investigation:** Kohei Ijichi.

**Methodology:** Reiko Tokuyama-Toda, Kazuhito Satomura.

**Resources:** Mai Takeshita-Umehara, Toshio Yudo, Yusuke Takebe.

**Supervision:** Reiko Tokuyama-Toda.

**Visualization:** Mai Takeshita-Umehara, Yusuke Takebe.

**Writing – original draft:** Kohei Ijichi, Reiko Tokuyama-Toda.

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
