## [Decision Letter · Decision Letter 0]

PONE-D-25-05017Development of a new method to promote tooth extraction socket healing through immediate transplantation of extracted dental pulp tissuePLOS ONE

Dear Dr. Tokuyama-Toda,

Thank you for submitting your manuscript to PLOS ONE. After careful consideration, we feel that it has merit but does not fully meet PLOS ONE’s publication criteria as it currently stands. Therefore, we invite you to submit a revised version of the manuscript that addresses the points raised during the review process.

We look forward to receiving your revised manuscript.

Kind regards,

Carlos Alberto Antunes Viegas, DVM; MSc; PhD

Academic Editor

PLOS ONE

3. To comply with PLOS ONE submissions requirements, in your Methods section, please provide additional information regarding the experiments involving animals and ensure you have included details on (1) methods of anesthesia and/or analgesia, and (3) efforts to alleviate suffering.

5. We note that you have indicated that there are restrictions to data sharing for this study. For studies involving human research participant data or other sensitive data, we encourage authors to share de-identified or anonymized data. However, when data cannot be publicly shared for ethical reasons, we allow authors to make their data sets available upon request. For information on unacceptable data access restrictions, please see http://journals.plos.org/plosone/s/data-availability#loc-unacceptable-data-access-restrictions.

6. We note that your Data Availability Statement is currently as follows: [All relevant data are within the manuscript and its Supporting Information files.]

7. We note that you have included the phrase “data not shown” in your manuscript. Unfortunately, this does not meet our data sharing requirements. PLOS does not permit references to inaccessible data. We require that authors provide all relevant data within the paper, Supporting Information files, or in an acceptable, public repository. Please add a citation to support this phrase or upload the data that corresponds with these findings to a stable repository (such as Figshare or Dryad) and provide and URLs, DOIs, or accession numbers that may be used to access these data. Or, if the data are not a core part of the research being presented in your study, we ask that you remove the phrase that refers to these data.

Reviewers' comments:

Reviewer's Responses to Questions

**Comments to the Author**

1. Is the manuscript technically sound, and do the data support the conclusions?

Reviewer #1: Yes

Reviewer #2: Partly

2. Has the statistical analysis been performed appropriately and rigorously? 

Reviewer #1: No

Reviewer #2: Yes

3. Have the authors made all data underlying the findings in their manuscript fully available?

Reviewer #1: Yes

Reviewer #2: Yes

4. Is the manuscript presented in an intelligible fashion and written in standard English?

Reviewer #1: Yes

Reviewer #2: Yes

5. Review Comments to the Author

Reviewer #1: Manuscript Title: Development of a new method to promote tooth extraction socket healing through immediate transplantation of extracted dental pulp tissue

Manuscript Number: PONE-D-25-05017

Journal: PLOS ONE

Comments to authors

Dear authors,

Thank you for submitting your manuscript for possible publication in PLOS ONE. This study explores a novel approach to promoting extraction socket healing through the immediate transplantation of dental pulp tissue. The topic is innovative and relevant to regenerative dentistry. However, major revisions are needed, particularly in methodological clarity and the presentation of results. Please find my detailed comments below.

Specific Comments

Abstract

● Avoid first-person language. Instead of “we investigated”, I suggest rephrasing as “this study aimed to investigate”. According to the journal’s guidelines, the abstract should explicitly describe the main objectives of the study. Please include this crucial information.

● Include numerical results.

Introduction

● Lines 45-49: For better readability, I suggest numbering the healing phases as (1); (2); etc.

● Line 51: Post-extraction discomfort is primarily linked to the inflammatory response rather than the complete formation of new bone. Please provide references to support the statement that discomfort persists until bone formation is complete or rephrase this paragraph.

● Line 52: "When a wisdom tooth is extracted, quality of life (QOL) is significantly reduced due to the location and size of the defective tooth." → Please provide references to support this claim.

● Line 53: This sentence is repetitive. I suggest removing it.

Methods

● Line 81: Specify the total number of rats used in the study.

● There is no information on the operators and data analyzers—were they calibrated? Were the procedures performed by experienced researchers?

● Was a sample size calculation performed to justify n = 5 at each time point?

● Provide references supporting the image measurement methodology—are there similar studies using this approach?

● When describing the data collection process, specify how animal suffering was minimized, including euthanasia and medications.

● Provide more details on how the radiopaque area was measured.

● The methodology section should explicitly describe how measurements were taken.

Results

● Report numerical results instead of general descriptions.

● Ensure that descriptions of measurement procedures appear only in the Methods section, not in the Results.

Discussion (Lines 145-147)

● The authors should emphasize their own results while comparing them to previous studies. For example, compare histological and imaging results with findings from other regenerative therapy investigations.

● Line 249: Instead of stating your future research plans, rephrase your sentences as suggestions for future studies.

Conclusion

● Line 267: The final sentence is unnecessary. Instead of stating future plans, rephrase them as suggestions for future research in the discussion section, particularly after stating the study limitations.

Additional Points

● The manuscript should be better structured to ensure information is appropriately categorized within the correct sections.

● The Results section should only present findings without interpretation—interpretative analysis belongs in the Discussion.

● Instead of stating "data not shown," consider creating a table to present numerical results, comparing both groups across different time points.

● Figure legends should be descriptive and concise, avoiding interpretation, which should be reserved for the Discussion.

Final Recommendation

The manuscript needs revisions to improve methodological clarity, structure, and overall writing. Major points to address include:

1. Providing missing numerical results in the Results section.

2. Revising the structure so that Methods, Results, and Discussion are properly divided.

○ According to the journal’s guidelines, separating Results, Discussion, and Conclusion is optional. However, since the authors have chosen to present them as distinct sections, each one should be written accordingly.

○ The Results section must present findings only, while the Discussion should summarize the main findings and compare them with existing literature.

3. Improving methodological transparency by detailing sample size calculations, operator calibration, and measurement protocols. The methods should be replicable.

4. Editing for clarity to ensure the writing is precise, concise, and professional.

Reviewer #2: Dear authors,

I evaluated the article titled "Development of a new method to promote tooth extraction socket healing through immediate transplantation of extracted dental pulp tissue”. Its goal was to invvestigate “the possibility of promoting tooth extraction socket healing by extracting dental pulp tissue from extracted teeth that would normally be discarded and transplanting it immediately into the socket.”

I considered that this study has no novelty. Therefore, it is interesting and can aggregate data to the literature. Moreover, it is contributing to the regenerative medicine area.

Comments per section:

TITLE: I suggest to include “An in vitro study”

ABSTRACT:

- remove personal pronouns

REFERENCES: this section is outdated. Only 24% of the references were published in the last 5 years. Update the literature (suggestions sent below)

INTRO

- for the first phrase (Ln.. 42-43) “Tooth extraction is one of the most prevalent surgical procedures in clinical oral surgery”, I suggest to include refs.

(new ref.) Effects of ozone therapy on postoperative pain, swelling, and trismus caused by surgical extraction of unerupted lower third molars: a double-blinded split-mouth randomized controlled trial. Med Oral Patol Oral Cir Bucal. 2025. doi:10.4317/medoral.26974

(new ref.) Evaluating the effectiveness of low-level laser therapy in patients undergoing lower third molar extraction: A double-blinded randomized controlled trial. Med Oral Patol Oral Cir Bucal. 2024. doi:10.4317/medoral.26894

- Ln 49-52: the refs. above can be used in these lines too.

- Ln 60-61: “and it has been reported that stem cells also exist in human dental pulp tissue [11-13].” I suggest to include the article below here.

(new ref.) The influence of Aloe vera with mesenchymal stem cells from dental pulp on bone regeneration: characterization and treatment of non-critical defects of the tibia in rats. Journal of Applied Oral Science, v. 27, p. 1-11, 2019. doi: 10.1590/1678-7757-2018-0103

M&M

- why the authors did not present a morphometrical analysis?

- how many authors participated of the image analysis?

RESULTS: there was subjective description of the histologic pieces analyzed. It could be presented with numbers to corroborate the micro-CT results.

DISCUSSION: please adjust the discussion present paragraphs.

Form Ln218 to Ln238, no one reference… part of the results section was presented. Add refs.

CONCLUSION: please revise the conclusion. The procedure is not simple and inexpensive.

Ln 265-266: The results suggest that immediate transplantation of dental pulp tissue into tooth extraction sockets can be a simple, safe, and inexpensive method

6. PLOS authors have the option to publish the peer review history of their article (what does this mean? ). If published, this will include your full peer review and any attached files.

**Do you want your identity to be public for this peer review?** For information about this choice, including consent withdrawal, please see our Privacy Policy .

Reviewer #1: **Yes: ** Gabriel Magrin

Reviewer #2: No

---

## [Author Response · Author response to Decision Letter 1]

11 Mar 2025

Mar 11, 2025

Dear Reviewers

PLOS ONE

Re: Manuscript Number: PONE-D-25-05017  

Title: Development of a new method to promote tooth extraction socket healing through immediate transplantation of extracted dental pulp tissue

Thank you for your valuable comments concerning our manuscript entitled "Development of a new method to promote tooth extraction socket healing through immediate transplantation of extracted dental pulp tissue".

We have carefully studied your comments and made the necessary corrections, and are sending here the revised manuscript again. The corrected document has colored text.

Our responses to your comments are as follows:

Response to the comments of Reviewer #1

< Specific Comments >

1. Abstract

Avoid first-person language. Instead of “we investigated”, I suggest rephrasing as “this study aimed to investigate”. According to the journal’s guidelines, the abstract should explicitly describe the main objectives of the study. Please include this crucial information.

Response

Thank you for your comment.

As you pointed out, I avoided first-person language and clearly stated the main objective.

“The main objective of this study was to verify whether immediate transplantation of dental pulp tissue without any culture process into the extraction socket would promote healing of the socket.” (line 22 to 24 on page 2, Revised Manuscript with Track Changes)

2. Include numerical results.

Response

Thank you for your comment.

I've added the numerical results as you suggested on line 34 and 37 on page 2, Revised Manuscript with Track Changes)

3. Introduction

Lines 45-49: For better readability, I suggest numbering the healing phases as (1); (2); etc.

Response

Thank you for your comment.

I've added the numbering on line 48 to 51 on page 4, Revised Manuscript with Track Changes.

4. Line 51: Post-extraction discomfort is primarily linked to the inflammatory response rather than the complete formation of new bone. Please provide references to support the statement that discomfort persists until bone formation is complete or rephrase this paragraph.

Response

Thank you for your valuable comment. As you pointed out, I changed the sentence to say that patients experience discomfort associated with an inflammatory response as follows.

“Immediately after tooth extraction and until a certain degree of healing has occurred, patients experience discomfort due to the inflammatory response.” (line 55 to 57 on page 4, Revised Manuscript with Track Changes)

5. Line 52: "When a wisdom tooth is extracted, quality of life (QOL) is significantly reduced due to the location and size of the defective tooth." → Please provide references to support this claim.

Response

Thank you for your comment.

I've added the references on line 58 on page 4, Revised Manuscript with Track Changes.

6. Line 53: This sentence is repetitive. I suggest removing it.

Response

Thank you for your comment.

As you pointed out, I removed the repetitive sentences and made them more concise on line 58 to 64 on page 4, Revised Manuscript with Track Changes.

7. Methods

Line 81: Specify the total number of rats used in the study.

Response

Thank you for your comment.

I've added total number of rats on line 96 to 97 on page 6, Revised Manuscript with Track Changes.

8. There is no information on the operators and data analyzers—were they calibrated? Were the procedures performed by experienced researchers?

Response

Thank you for your valuable comments.

All procedures in this experiment were performed by one operator, and data analysis was performed by three researchers. The details have been added as follows.

“These procedures were performed by a single experienced operator under the guidance of an oral surgeon certified by the Japanese Society of Oral and Maxillofacial Surgeons. The image analysis was conducted by three authors (the corresponding author, the first author, and one other person). (line 108 to 112 on page 6, Revised Manuscript with Track Changes)

9. Was a sample size calculation performed to justify n = 5 at each time point?

Response

Thank you for your valuable comments.

The sample size determination was detailed as follows:

“The sample number was elected by power test analysis. Level of significance of 5% and power test of 95% were adopted, and it was suggested four animals per group. Thus, with a possible animal loss, it was used five per period of analysis.” (line 90 to 93 on page 5 to 6, Revised Manuscript with Track Changes)

10. Provide references supporting the image measurement methodology—are there similar studies using this approach?

Response

Thank you for your comment.

The measurements were made in a modified form from previous studies. I have detailed the measurements as follows, and added figure 1 to clarify the measurement method.

“A previously reported method [29-31] was modified to determine radiolucent and radiopaque areas in tooth extraction sockets. That is, due to the irregular alveolar socket morphology, the region of interest (ROI) was standardized as a cube enclosed by the determined plane, which based on a straight line passing through the center of the distal buccal roots of the bilateral maxillary first molars and the tooth axis of the distal buccal roots of the bilateral maxillary first molars. The above plane was determined on the 5th day when the radiolucent area expanded due to bone resorption caused by inflammation after tooth extraction, and the range of the extraction socket to be measured was determined to be the range from 5-10 sections to 85-90 sections from the set plane (Fig 1).” (line 125 to 135 on page 7, Revised Manuscript with Track Changes)

11. When describing the data collection process, specify how animal suffering was minimized, including euthanasia and medications.

Response

Thank you for your comment.

I have added the following sentence regarding anesthesia methods and euthanasia in animal experiments:

“under a combination anesthetic (0.375 mg/kg of medetomidine, 2.0 mg/kg of midazolam, and 2.5 mg/kg of butorphanol)” (line 99 to 100 on page 6, Revised Manuscript with Track Changes)

“25 animals (n = 5 per group) were euthanized by anesthesia overdose (pentobarbital sodium, 100 mg/kg).” (line 158 to 159 on page 8, Revised Manuscript with Track Changes)

12. Provide more details on how the radiopaque area was measured.

Response

Thank you for your valuable comments.

I have added references based on your comments in the previous section and have completely rewritten the method for measuring radiopaque areas using μCT. (line 115 to 151 on page 7 to 8, Revised Manuscript with Track Changes)

13. The methodology section should explicitly describe how measurements were taken.

Response

Thank you for your detailed explanation. As shown in the previous section, I have clarified the measurement method according to your explanation. (line 115 to 151 on page 7 to 8, Revised Manuscript with Track Changes)

14. Results

Report numerical results instead of general descriptions.

Response

Thank you for your comment.

I've added numerical results on line 191 to 192, 194, 197, 198 on page 10, line 213 to 214, 215 to 216, 226 to 228 on page 11, line 232, 233, 238 on page 12, Revised Manuscript with Track Changes.

15. Ensure that descriptions of measurement procedures appear only in the Methods section, not in the Results.

Response

Thank you for your comment.

I've removed descriptions of measurement procedures in the Results on line 175 to 176, on page 9, line 183 to 187 on page 10, line 203 to 207 on page 10 to 11, Revised Manuscript with Track Changes.

16. Discussion (Lines 145-147)

The authors should emphasize their own results while comparing them to previous studies. For example, compare histological and imaging results with findings from other regenerative therapy investigations.

Response

Thank you for your valuable comments.

I have considered our findings in reference to the results of previous studies and added the following to the manuscript.

“The alveolar bone healing in the control group supported previous studies [29, 32-35], and there were no problems with the healing process in this study. Based on these findings, direct transplantation of extracted dental pulp tissue may promote healing of the extraction socket.” (line 294 to 298 on page 14, Revised Manuscript with Track Changes)

“Previous studies have shown that prolonged inflammation after tooth extraction delays the healing of the alveolar bone[36-39], and these findings suggest that earlier suppression of the inflammatory response will lead to promoted healing of the tooth extraction socket.” (line 304 to 307 on page 15, Revised Manuscript with Track Changes)

“Previous reports have shown that applying bio-cement to the extraction socket and maintaining the alveolar crest allows for uniform healing of the alveolar bone [40]. In present study, the transplanted dental pulp tissue also promoted new bone formation near the alveolar crest, which may have contributed to the promotion of healing.” (line 312 to 316 on page 15, Revised Manuscript with Track Changes)

“Numerous studies have been reported on promoting healing in tooth extraction sockets, whether the subjects were in a normal or delayed healing condition [1-3, 30, 37, 39-42]. Furthermore, many studies have been conducted using MSCs, including DPSCs [18, 43-47]” (line 322 to 325 on page 15 to 16, Revised Manuscript with Track Changes)

17. Line 249: Instead of stating your future research plans, rephrase your sentences as suggestions for future studies.

Response

Thank you for your comment.

I've changed sentence “As suggestions for future studies” instead of “In the future” on line 334 on page 16, Revised Manuscript with Track Changes.

18. Conclusion

Line 267: The final sentence is unnecessary. Instead of stating future plans, rephrase them as suggestions for future research in the discussion section, particularly after stating the study limitations.

Response

Thank you for your comment.

I've removed final sentence on line 359 to 361 on page 17, Revised Manuscript with Track Changes.

< Additional Points >

1. The manuscript should be better structured to ensure information is appropriately categorized within the correct sections.

Response

Thank you for your careful review.

I have edited the manuscript to ensure that statements are placed in the appropriate sections throughout.

2. The Results section should only present findings without interpretation—interpretative analysis belongs in the Discussion.

Response

Thank you for your comment.

I've removed sentence about interpretative findings on line 199 to 203 on page 10, Revised Manuscript with Track Changes.

3. Instead of stating "data not shown," consider creating a table to present numerical results, comparing both groups across different time points.

Response

Thank you for your comment.

I have added all the data that was marked as "data not shown." Therefore, I have edited revised figure 2 and revised figure 3.

4. Figure legends should be descriptive and concise, avoiding interpretation, which should be reserved for the Discussion.

Response

Thank you for your comment.

I've avoided sentence about interpretative findings on line 221 to 224 on page 11, line 253 to 254 on page 13, Revised Manuscript with Track Changes.

< Final Recommendation >

The manuscript needs revisions to improve methodological clarity, structure, and overall writing. Major points to address include:

1. Providing missing numerical results in the Results section.

Response

Thank you for your careful review.

As I said in response to the previous comment, I added numerical results in the Results section.

2. Revising the structure so that Methods, Results, and Discussion are properly divided.

According to the journal’s guidelines, separating Results, Discussion, and Conclusion is optional. However, since the authors have chosen to present them as distinct sections, each one should be written accordingly.

The Results section must present findings only, while the Discussion should summarize the main findings and compare them with existing literature.

Response

Thank you for your careful review.

As I said in response to the previous comment, I have edited the manuscript to ensure that statements are placed in the appropriate sections throughout.

3. Improving methodological transparency by detailing sample size calculations, operator calibration, and measurement protocols. The methods should be replicable.

Response

Thank you for your careful review.

As I said in response to the previous comment, following your suggestions, I have significantly revised the manuscript to make the methodology clearer.

4. Editing for clarity to ensure the writing is precise, concise, and professional.

Response

Thank you for your careful review.

I have edited the manuscript following your suggestions to make it more concise and clear.

In addition, based on comments from other reviewers, I added some sentences.

Response to the comments of Reviewer #2

1. TITLE: I suggest to include “An in vitro study”

Response

Thank you for your comment.

As you pointed out, I added “ An in vivo study” in title. (line 2 on page 1, Revised Manuscript with Track Changes)

2. ABSTRACT:

- remove personal pronouns

Response

Thank you for your comment.

As you pointed out, I avoided first-person language through the manuscript.

3. REFERENCES: this section is outdated. Only 24% of the references were published in the last 5 years. Update the literature (suggestions sent below)

INTRO

- for the first phrase (Ln.. 42-43) “Tooth extraction is one of the most prevalent surgical procedures in clinical oral surgery”, I suggest to include refs.

(new ref.) Effects of ozone therapy on postoperative pain, swelling, and trismus caused by surgical extraction of unerupted lower third molars: a double-blinded split-mouth randomized controlled trial. Med Oral Patol Oral Cir Bucal. 2025. doi:10.4317/medoral.26974

(new ref.) Evaluating the effectiveness of low-level laser therapy in patients undergoing lower third molar extraction: A double-blinded randomized controlled trial. Med Oral Patol Oral Cir Bucal. 2024. doi:10.4317/medoral.26894

- Ln 49-52: the refs. above can be used in these lines too.

- Ln 60-61: “and it has been reported that stem cells also exist in human dental pulp tissue [11-13].” I suggest to include the article below here.

(new ref.) The influence of Aloe vera with mesenchymal stem cells from dental pulp on bone regeneration: characterization and treatment of non-critical defects of the tibia in rats. Journal of Applied Oral Science, v. 27, p. 1-11, 2019. doi: 10.1590/1678-7757-2018-0103

Response

Thank you for your careful review.

I have edited the manuscript to include all the references you suggested as indicated and have added other new references as necessary.

(line 47, 58, 70 in Introduction section on page 4, Revised Manuscript with Track Changes)

4. M&M

- why the authors did not present a morphometrical analysis?

- how many authors participated of the image analysis?

Response

Thank you for your valuable comments.

I have added more details about the morphometrical analysis to the manuscript and added references. I have also added figure 1 to clarify the measurement method.

“A high-resolution X-ray CT (inspXio SMX-225CT, Shimadzu, Kyoto, Japan) was used to non-destructively observe the internal structure of maxilla of five Sprague Dawley�SD�rat at each scheduled day. The imaging conditions were as follows: tube voltage, 110 kV; tube current, 70 µA; resolution, 1024 × 1024 pixels; number of scanning views, 1200, and total scanning time, 900 s. Image processing software (TRI/3D-BON-FCS64, RATOC, Tokyo, Japan) was used for 3D image analysis. A previously reported method [29-31] was modified to determine radiolucent and radiopaque areas in tooth extraction sockets. That is, due to the irregular alveolar socket morphology, the region of interest (ROI) was standardized as a cube enclosed by the determined plane, which based on a straight line pas

---

## [Decision Letter · Decision Letter 1]

PONE-D-25-05017R1Development of a new method to promote tooth extraction socket healing through immediate transplantation of extracted dental pulp tissuePLOS ONE

Dear Dr. Tokuyama-Toda,

Thank you for submitting your manuscript to PLOS ONE. After careful consideration, we feel that it has merit but does not fully meet PLOS ONE’s publication criteria as it currently stands. Therefore, we invite you to submit a revised version of the manuscript that addresses the points raised during the review process.

We look forward to receiving your revised manuscript.

Kind regards,

Carlos Alberto Antunes Viegas, DVM; MSc; PhD

Academic Editor

PLOS ONE

Journal Requirements:

Reviewers' comments:

Reviewer's Responses to Questions

**Comments to the Author**

1. If the authors have adequately addressed your comments raised in a previous round of review and you feel that this manuscript is now acceptable for publication, you may indicate that here to bypass the “Comments to the Author” section, enter your conflict of interest statement in the “Confidential to Editor” section, and submit your "Accept" recommendation.

Reviewer #1: All comments have been addressed

Reviewer #2: All comments have been addressed

2. Is the manuscript technically sound, and do the data support the conclusions?

Reviewer #1: Partly

Reviewer #2: Yes

3. Has the statistical analysis been performed appropriately and rigorously? 

Reviewer #1: Yes

Reviewer #2: Yes

4. Have the authors made all data underlying the findings in their manuscript fully available?

Reviewer #1: No

Reviewer #2: Yes

5. Is the manuscript presented in an intelligible fashion and written in standard English?

Reviewer #1: Yes

Reviewer #2: Yes

6. Review Comments to the Author

Reviewer #1: Manuscript Title: Development of a new method to promote tooth extraction socket healing through immediate transplantation of extracted dental pulp tissue

Manuscript Number: PONE-D-25-05017

Journal: PLOS ONE

Dear Authors,

Thank you for carefully considering my previous comments and suggestions. Your manuscript has improved significantly. Below, I provide additional comments for further refinement.

Abstract

● Lines 22-27: Some parts remain repetitive; consider revising for conciseness.

● When stating your results, clarify whether the observed increases were significantly different between groups.

Materials and Methods

● Line 119: "Two groups based on variations in the radiopaque area." Please specify the unit of measurement for clarity.

● Mentioning the third author: If identifying them in the text, please do so by initials rather than by general mention.

Results

● I still believe it would be beneficial to include a table presenting the numerical results (volume of the radiopaque area and rate of change) for both groups across different time points. Additionally, indicate whether the differences were statistically significant.

● Lines 188-190, 210: Past tense is more appropriate for the Methods section; consider rephrasing. For example, in the Results section, you can say:

"Changes in the radiopaque area compared between the two groups are presented in Fig. 2."

Discussion

● Line 321: You mention that "Numerous studies have been reported on promoting healing in tooth extraction." Have any of these studies used micro-CT and histology to evaluate assisted socket healing? If so, how do their results compare to yours? This would also be an opportunity to compare your numerical results with similar studies.

● Lines 345, 350: Avoid using first-person language; revise accordingly.

Conclusion

● The conclusion should answer the study’s main objective rather than introduce new information, similar to your conclusion in the abstract.

● Lines 355-356: "Compared to conventional regenerative medicine using stem cells obtained through..." This statement is not supported by your study, as you did not compare pulp transplantation to other regenerative therapies. Consider rewording or removing this part.

Reviewer #2: Dear authors,

Thank you so much for your responses and adjustments.

Congratulations from my end.

Best regards.

7. PLOS authors have the option to publish the peer review history of their article (what does this mean? ). If published, this will include your full peer review and any attached files.

**Do you want your identity to be public for this peer review?** For information about this choice, including consent withdrawal, please see our Privacy Policy .

Reviewer #1: **Yes: ** Gabriel Magrin

Reviewer #2: No

---

## [Author Response · Author response to Decision Letter 2]

15 Apr 2025

Mar 25, 2025

Dear Reviewers

PLOS ONE

Re: Manuscript Number: PONE-D-25-05017R1  

Title: Development of a new method to promote tooth extraction socket healing through immediate transplantation of extracted dental pulp tissue

Thank you for your valuable comments concerning our manuscript entitled "Development of a new method to promote tooth extraction socket healing through immediate transplantation of extracted dental pulp tissue".

We have carefully studied your comments and made the necessary corrections, and are sending here the revised manuscript again. The corrected document has colored text (blue text and light blue highlights).

Our responses to your comments are as follows:

Response to the comments of Reviewer #1

1. Abstract

Lines 22-27: Some parts remain repetitive; consider revising for conciseness.

Response

Thank you for your comment.

As you pointed out, I have removed the light blue highlighted parts and revised it to make it more concise. (line 24 to 27 on page 2, Revised Manuscript with Track Changes)

2. When stating your results, clarify whether the observed increases were significantly different between groups.

Response

Thank you for your comment.

As you pointed out, The results have been revised to include significant differences.　 (line 34 to 40 on page 2, Revised Manuscript with Track Changes)

3. Materials and Methods

Line 119: "Two groups based on variations in the radiopaque area." Please specify the unit of measurement for clarity.

Response

Thank you for your comment.

As you pointed out, I added the unit to the end of the sentence.

(line 121 on page 7, Revised Manuscript with Track Changes)

4. Mentioning the third author: If identifying them in the text, please do so by initials rather than by general mention.

Response

Thank you for your comment.

As you pointed out, The initials of the person conducting this study have been added.

(line 111 to 114 on page 6, Revised Manuscript with Track Changes)

5. Results

I still believe it would be beneficial to include a table presenting the numerical results (volume of the radiopaque area and rate of change) for both groups across different time points. Additionally, indicate whether the differences were statistically significant.

Response

Thank you for your comment.

As you pointed out, All the numerical results have been tabulated and added as Table 1.

(line 192 to 193 on page 10, and Table 1 (line 237 to 240 on page 12), Revised Manuscript with Track Changes)

6. Lines 188-190, 210: Past tense is more appropriate for the Methods section; consider rephrasing. For example, in the Results section, you can say:

"Changes in the radiopaque area compared between the two groups are presented in Fig. 2."

Response

Thank you for your comment.

As you pointed out, I have revised the past tense and rephrased it as per your instructions.

(line 190 to 193 on page 10, line 213 to 214 on page 11, Revised Manuscript with Track Changes)

7. Discussion

Line 321: You mention that "Numerous studies have been reported on promoting healing in tooth extraction." Have any of these studies used micro-CT and histology to evaluate assisted socket healing? If so, how do their results compare to yours? This would also be an opportunity to compare your numerical results with similar studies.

Response

Thank you for your comment.

As you pointed out, I have added a sentence comparing the results of this study with the results of previous studies.

(line 331 to 341 on page 16, Revised Manuscript with Track Changes)

8. Lines 345, 350: Avoid using first-person language; revise accordingly.

Response

Thank you for your comment.

As you pointed out, I avoided first-person language.

(line 363 on page 17 and 369 on page 18, Revised Manuscript with Track Changes)

9. Conclusion

The conclusion should answer the study’s main objective rather than introduce new information, similar to your conclusion in the abstract.

Response

Thank you for your comment.

As you pointed out, I have only stated the conclusion briefly.

(line 371 to 373 on page 18, Revised Manuscript with Track Changes)

10. Lines 355-356: "Compared to conventional regenerative medicine using stem cells obtained through..." This statement is not supported by your study, as you did not compare pulp transplantation to other regenerative therapies. Consider rewording or removing this part.

Response

Thank you for your comment.

As you pointed out, I removed the part you pointed out.

(line 373 to 376 on page 18, Revised Manuscript with Track Changes)

Response to the comments of Reviewer #2

Thank you for reviewing our manuscript and giving us valuable feedback.

In addition, based on comments from other reviewers, I added some sentences.

We believe the manuscript has been improved satisfactorily and hope that it is now acceptable for publication in PLOS ONE.

Yours sincerely,

Reiko Tokuyama-Toda, DDS, PhD

---

## [Decision Letter · Decision Letter 2]

Development of a new method to promote tooth extraction socket healing through immediate transplantation of extracted dental pulp tissue

PONE-D-25-05017R2

Dear Dr. Reiko Tokuyama-Toda,

We’re pleased to inform you that your manuscript has been judged scientifically suitable for publication and will be formally accepted for publication once it meets all outstanding technical requirements.

Kind regards,

Carlos Alberto Antunes Viegas, DVM; MSc; PhD

Academic Editor

PLOS ONE

Additional Editor Comments (optional):

Reviewers' comments:

Reviewer's Responses to Questions

**Comments to the Author**

1. If the authors have adequately addressed your comments raised in a previous round of review and you feel that this manuscript is now acceptable for publication, you may indicate that here to bypass the “Comments to the Author” section, enter your conflict of interest statement in the “Confidential to Editor” section, and submit your "Accept" recommendation.

Reviewer #1: All comments have been addressed

2. Is the manuscript technically sound, and do the data support the conclusions?

Reviewer #1: Yes

3. Has the statistical analysis been performed appropriately and rigorously? 

Reviewer #1: Yes

4. Have the authors made all data underlying the findings in their manuscript fully available?

Reviewer #1: Yes

5. Is the manuscript presented in an intelligible fashion and written in standard English?

Reviewer #1: Yes

6. Review Comments to the Author

Reviewer #1: Dear authors,

I would like to thank the authors for addressing the suggestions and comments provided in the previous review. After reading the revised version of the manuscript, I am satisfied that the changes have been implemented, significantly improving the quality of the work. In its current form, I consider the manuscript suitable for publication.

7. PLOS authors have the option to publish the peer review history of their article (what does this mean? ). If published, this will include your full peer review and any attached files.

**Do you want your identity to be public for this peer review?** For information about this choice, including consent withdrawal, please see our Privacy Policy .

Reviewer #1: **Yes: ** Gabriel Magrin

---

## [Editor Report · Acceptance letter]

PONE-D-25-05017R2

PLOS ONE

Dear Dr. Tokuyama-Toda,

I'm pleased to inform you that your manuscript has been deemed suitable for publication in PLOS ONE. Congratulations! Your manuscript is now being handed over to our production team.

Kind regards,

on behalf of

Dr. Carlos Alberto Antunes Viegas

Academic Editor

PLOS ONE